# Advanced Bladder Cancer: Changing the Treatment Landscape

**DOI:** 10.3390/jpm12101745

**Published:** 2022-10-20

**Authors:** Vladimir Bilim, Hiroo Kuroki, Yuko Shirono, Masaki Murata, Kaede Hiruma, Yoshihiko Tomita

**Affiliations:** 1Department of Urology, Division of Molecular Oncology, Graduate School of Medical and Dental Sciences, Niigata University, 1-757 Asahimachi-dori, Chuo-ku, Niigata 951-8510, Japan; 2Kameda Daiichi Hospital, Niigata 950-0165, Japan; 3Nagaoka Red Cross Hospital, Senshu-2 297-1, Nagaoka-shi 940-2085, Japan

**Keywords:** urothelial carcinoma, bladder cancer

## Abstract

Bladder cancer is the 10th most common cancer type in the world. There were more than 573,000 new cases of bladder cancer in 2020. It is the 13th most common cause of cancer death with an estimated more than 212,000 deaths worldwide. Low-grade non-muscle-invasive bladder cancer (NMIBC) is usually successfully managed with transurethral resection (TUR) and overall survival for NMIBC reaches 90% according to some reports. However, long-term survival for muscle-invasive bladder cancer (MIBC) and metastatic bladder cancer remains low. Treatment options for bladder cancer have undergone a rapid change in recent years. Immune checkpoint inhibitors (ICI), targeted therapies, and antibody-drug conjugates are available now. As bladder cancer is genetically heterogeneous, the optimization of patient selection to identify those most likely to benefit from a specific therapy is an urgent issue in the treatment of patients with bladder cancer.

## 1. Introduction

Bladder cancer (BC) is the 10th most common cancer type in the world [1]. There were more than 573,000 new cases of bladder cancer in 2020. It is the 13th most common cause of cancer death with an estimated more than 212,000 deaths worldwide. BC is the fourth most prevalent malignancy among American males and the eighth most common cause of cancer death [1]. Bladder cancer is the second most common malignancy involving the urinary system following prostate cancer.

There are three main histological types of BC: transitional cell carcinoma (urothelial carcinoma), which accounts for 90%; squamous cell carcinoma, which accounts for about 5%; and adenocarcinoma, which presents less than 2% of all BCs [2]. Other rare histological types of BC comprise a small percent.

Tobacco smoking is one of the established BC risk factors. Many chemicals have been also associated with BC risk. Aromatic amines including aniline are identified chemical causes of BC. Arsenic in drinking water has been associated with bladder cancer in several geographic regions. Radiotherapy for pelvic organs (prostate, rectum, and uterus) has been demonstrated to increase the risk of BC [3,4]. Chronic bladder irritation and infection such as the presence of bladder stones, continuously indwelled bladder catheter, and schistosomiasis (mainly in Africa and the Middle East) have been associated with the increased risk of bladder squamous cell cancer. Urachal carcinoma is non-urothelial bladder cancer. It is an adenocarcinoma arising in the urachal remnant [5].

Upper tract urothelial carcinoma (UTUC) is histologically similar to urothelial BC. However, UTUC has several unique clinical, biological, and molecular features [6]. Compared to the occurrence of BC, primary upper tract (renal calyces, renal pelvis, and ureter) urothelial carcinoma (UC) is a relatively rare tumor: it accounts for only 5% to 10% of all urothelial carcinomas [7]. Although upper urinary tract and bladder UC share many common biological pathways, the upper urinary tract mutational signatures differ, and four unique molecular and clinical subtypes were identified in the upper urinary tract UC using clustering of RNA expression data [8]. These clusters were similar to the previously described bladder subtypes [9], but had unique features compared to bladder cancer. Cluster 1 was similar to the bladder luminal subtype, while cluster 2 was more similar to the basal subtype and had worse overall survival. Both clusters 2 and 3 had 100% FGFR3 mutations. At the same time, cluster 2 patients had no bladder recurrences, while cluster 3 patients had high rates of bladder recurrence and no TP53 mutations. Cluster 4 had a high rate of upregulated immune checkpoint genes, and it was enriched for high-grade, muscle-invasive disease. In spite of the differences between bladder UC and UTUC, treatment guidelines for UTUC are generally based on evidence that is based on the data obtained for bladder UC [6].

Novel classification based on genetic data has identified five unique molecular subtypes of BC: luminal-papillary (35%), luminal-infiltrated (19%), luminal (6%), basal SCC-like (35%), and neuronal-like (5%) [9]. At present, a different approach is recommended only for neuroendocrine tumors. Because of the high rate of these tumors, an aggressive approach is recommended. Neoadjuvant chemotherapy with a cystectomy is generally offered for patients [10]. In patients with organ-preserving treatment, combined radio-chemotherapy is performed. The chemotherapy regimen includes etoposide and cisplatin. The high mutational profile of basal tumors might explain their response to immune-oncology drugs. The basal tumors also demonstrate a high degree of mutations in the erbB family, and the treatment with an EGFR inhibitor might give an advantage in this tumor subtype [11]. Basal tumors have the largest survival benefit from cisplatin-based neoadjuvant chemotherapy [12], and also these tumors responded best to nivolumab [13]. Luminal-papillary tumors bear FGFR3 mutations [14] and are susceptible to FGFR3 kinase inhibitors.

About three-fourths of patients with BC present with a non-muscle-invasive BC (NMIBC) at the first visit. Ten to fifteen percent of patients with muscle-invasive BC (MIBC) have metastasis at the time of diagnosis [15]; and about half of MIBC patients treated with radical intent by cystectomy will relapse. One-third of them will have a local recurrence and the remnant 70% will have distant metastases. About 25% of newly diagnosed patients with bladder cancer have muscle-invasive bladder cancer (MIBC) or metastatic disease [16]. Despite the currently available multimodality therapy, advanced bladder cancer has a very low overall survival (OS) rate. For MIBC (T2), the 5-year survival rate is 70%. About 33% of bladder cancers are diagnosed at this stage. For T3 and above or N1 and above BC, the 5-year survival rate is 38%. For metastatic BC, the 5-year survival rate is about 6% [17].

The standard of care treatment for advanced BC (local invasive and/or metastatic) was cytotoxic therapy. Platinum-based cytotoxic chemotherapy is a standard 1st-line treatment for advanced BC (Figure 1). Since 1980, a combination of methotrexate, vinblastine, doxorubicin, and cisplatin (MVAC) has been a standard chemotherapy regimen [18]. The median survival time of patients with metastatic disease treated with MVAC was only 13 months [18,19]. Since the late 1990s, a gemcitabine and cisplatin (GC) combination has been established [20]. GC therapy did not demonstrate an advantage in terms of survival but was less toxic than MVAC. A dose-dense (dd) MVAC pathological complete response and overall response with dd-MVAC were not significantly different from the classic MVAC, but overall survival was significantly improved [21]. Salvage treatment for metastatic disease includes ICI, ADCs (antibody-drug conjugate), and FGFR inhibitors.

## 2. Immune Checkpoint Inhibitors

Clinical trials studying immune checkpoint inhibitors (ICI) (Figure 1, Table 1 and Table 2) have demonstrated prolonged OS and improved quality of life, and several ICIs are approved for routine clinical use in advanced BC (Table 2). Taking into account ICIs’ safety profile, they can be used in elderly or cisplatin-unfit patients.

The upfront standard therapy for metastatic UC includes platinum-based therapy, gemcitabine and cisplatin (for cisplatin-eligible patients), or gemcitabine and carboplatin (for cisplatin-ineligible patients) chemotherapy followed by maintenance avelumab (Figure 1, Table 3). In cisplatin-ineligible patients with PD-L1 positive tumors, upfront pembrolizumab has been approved (Figure 1, Table 3). In a KEYNOTE 045 trial for pembrolizumab in patients with metastatic or locally advanced/unresectable UC that has recurred or progressed following platinum-based chemotherapy, the subgroup analysis of PD-L1 positive tumors (combined positive score (CPS), positive tumor cells, lymphocytes, and macrophages ≥ 1%) and strongly PD-L1 positive tumors (CPS ≥ 10%) was performed. In a long (more than 2 years) follow-up, pembrolizumab demonstrated an OS benefit over chemotherapy in all subgroups with different levels of PD-L1 expression (i.e., CPS < 1, CPS ≥ 1, CPS < 10, and CPS ≥ 10) [22]. Thus, based on these results, PD-L1 expression is not a good predictive tool of ICI response in BC. However, using the tumor proportion score (TPS) might be a better predictor for ICIs’ efficacy. On the other hand, both CPS and TPS proved to be equally predictive of response to anti-PD-1/PD-L1 therapy in non-small cell lung cancer [23].

## 3. Antibody-Drug Conjugates

Antibody-drug conjugates (ADCs) are now established as a novel class of therapeutics for the treatment of cancer. In ADCs, monoclonal antibodies, specifically targeting tumor cells, are conjugated to a small molecule chemotherapeutic (payload) that fails to demonstrate sufficient efficacy on its own at tolerable doses due to high toxicity. The antibody binds to the target antigen on the surface of a tumor cell and the whole conjugate is internalized. The payload is bound to an antibody via a cleavable crosslinker. Depending on the linker’s structure, it is cleaved by hydrolytic enzymes, intracellular reducing molecules, or a change of pH (endosomal pH = 5–6; lysosomal pH = 4.8). Thus, a chemotherapeutic agent is selectively delivered to cancer cells [24].

Nectin-4, a transmembrane protein, has been found to be highly expressed in BC. Eighty-three percent of all BC samples and 92% of metastatic BCs were positive for nectin-4 by immunohistochemical staining. Moderate to strong staining was especially observed in 64% of BCs [25]. In Enfortumab Vedotin, the nectin-4 targeting antibody is linked to the microtubule disrupting agent, monomethyl auristatin E (MMAE), via a protease-cleavable crosslinker. Auristatin E causes G2/M cell cycle arrest and apoptosis.

In an EV-201 trial (Table 2, Figure 1), patients with locally advanced or metastatic UC who had previously been treated with a PD-1/PD-L1 inhibitor were included. Those who received prior treatment with platinum-containing chemotherapy were in cohort 1, and those who had no platinum-containing chemotherapy were in cohort 2. All patient tumors evaluated were positive for Nectin-4 and all of them had strong expression. Cohort 1’s median follow-up time was 10.15 months (range, 0.49 to 16.46 months). Cohort 2’s median follow-up time was 13.4 months (range, 0.33 to 29.27 months). ORR was 32.2% in cohort 1, and 58% in cohort 2. The duration of objective response was 7.6 and 10.9 months in cohort 1 and cohort 2, respectively. In cohort 1, 12% had complete responses. Similar responses were observed in patients with no response to prior anti-PD-1/L1 therapy (https://clinicaltrials.gov/ct2/show/NCT03219333, accessed on 8 July 2022) [26]. The trial demonstrated that Enfortumab Vedotin was efficient in advanced BT that progressed after platinum-based therapy and ICIs.

In December 2019, Enfortumab Vedotin gained breakthrough therapy designation by the FDA for patients with locally advanced or metastatic UC who previously received ICIs (Figure 1, Table 3).

Trop-2 is an epithelial glycoprotein that is differentially expressed in normal urothelium, non-invasive BC, and invasive BC tissues [27]. Trop-2 is overexpressed on the surface of invasive BC cells. In Sacituzumab Govitecan, an anti Trop-2 antibody is bound to the topoisomerase I inhibitor SN-38. The linker has a site for cleavage by lysosomal enzymes and also contains a hydrolyzable carbonate moiety [28]. Sacituzumab Govitecan was evaluated in the TROPHY-U-01 phase II second line trial [29] (Table 2, Figure 1). A median follow-up was 9.1 months, and the ORR was 27%. The median duration of response was 7.2 months. There was no requirement for tumor Trop-2 expression for enrollment and, thus, it is impossible to discuss any correlation between patient responses and Trop-2 expression levels. In April 2021, the FDA granted accelerated approval to Sacituzumab Govitecan for patients with locally advanced or metastatic UC who have previously received a platinum-containing chemotherapy and a PD-1/PD-L1 inhibitor. Sacituzumab Govitecan has become the second antibody-drug conjugate approved for the treatment of BC (Figure 1, Table 3).

## 4. New Options for the High-Risk NMIBC

The standard care treatment for high-risk NMIBC (CIS, high-grade Ta) is a transurethral resection of the bladder tumor (TURBT) followed by intravesical Bacillus Calmette–Guerin (BCG) instillations with induction and maintenance therapy for up to 3 years [30]. BCG is used to treat unresectable CIS and prevent recurrence in TUR-resected high-grade NMIBC. Although the efficacy of BCG is up to 80%, 20% of CIS patients do not respond to BCG and more than half of responders recur, with the majority recurring within 1 year. Until recently, there were two options for these patients: radical cystectomy and cytotoxic chemotherapy. It has been demonstrated that PD-L1 and PD-1 expression was increased in the majority of BCG-treated samples [31]. These data provide a rationale for the application of ICIs for BCG non-responding patients. Recently, based on the results of the KEYNOTE-057 phase II trial [32], pembrolizumab has been FDA approved for the treatment of BCG-unresponsive patients who did not opt for, or were ineligible for, radical cystectomy. Pembrolizumab demonstrated a 40.6% complete response, with a median duration of response of 16.2 months.

There are several ongoing trials. The KEYNOTE-676 [33] is an ongoing open-label phase III study randomizing patients to receive either pembrolizumab and BCG versus BCG alone in high-risk NMIBC. The primary endpoint is the CCR rate in patients with CIS. The SWOG S1605 phase II trial preliminary data demonstrated that atezolizumab had a 41.1% complete response at 3 months. The PREVERT phase II study tested Avelumab. The CheckMate 9UT studied nivolumab, and durvalumab is being studied in the ADAPT-BLADDER study [34].

## 5. New Options for MIBC

Adjuvant nivolumab after radical cystectomy (CheckMate 274 trial) increased median disease-free survival (20.8 vs. 10.8 months with placebo), and a more impressive increase in median disease-free survival was observed in PD-L1-positive (PD-L1 ≥ 1%) patients (NR vs. 10.8 months) [35].

The Enfortumab Vedotin combination with pembrolizumab in untreated advanced UC in EV-302 EV-103 studies demonstrated an overall response of 73.3%, 15.6% complete response, and median progression-free survival of 12.3 months [36]. KEYNOTE-B15/EV-304 [37] is an ongoing phase III randomized open-label trial of Enfortumab Vedotin with pembrolizumab in cisplatin-eligible MIBC patients.

CTLA-4-disrupted cytotoxic T lymphocytes exhibited a pronounced anti-tumor effect in vivo in the subcutaneous xenograft BC model [38], making a rationale for CTLA-4 targeting in clinical trials. In CheckMate 032, an open-label study, previously treated patients with unresectable locally advanced or metastatic UC were treated with either nivolumab alone or one of two nivolumab plus ipilimumab combination regimens [39]. The analysis demonstrated that one of the combination regimens (NIVO1+IPI3) provided the greatest antitumor activity, with a manageable safety profile. In the NABUCCO trial, 24 patients with stage III UC received two doses of ipilimumab and two doses of nivolumab, followed by RC. Forty-six percent of the patients had a pathological CR, and 58% had downstaging to NMIBC [40]. DUTRENEO is another ongoing trial of a combination of an anti-PDL-1 (Durvalumab) + anti-CTLA4 (TREmelimumab) as a neoadjuvant approach (https://www.clinicaltrials.gov/ct2/show/NCT03472274, accessed on 8 July 2022). Another combination study of anti-PD-L1 (durvalumab) plus anti-CTLA-4 (tremelimumab) in a small group of 28 high-risk BC cisplatin-ineligible patients (NCT02812420) demonstrated a pathological complete response of 37.5% and downstaging to pT1 or less in 58% of patients [41], making a combination of PD-L1 and CTLA-4 blockade a promising approach in BC.

PrE0807 studied the combination of anti-PD-1 nivolumab and lirilumab (anti-killer-cell immunoglobulin-like receptor (KIR)) [42].

## 6. Old and Frail Patients

BC risk factors include advanced age, and the risk of BC increases with age. About 70% of BC patients are older than 65 years old. About 9 out of 10 patients with BC are older than 55 [43]. The average age people are diagnosed with bladder cancer is 73 [1]. Life expectancy is increasing globally. The incidence of frailty is increasing with age. It has been reported that frailty was related to treatment modality selection in patients with muscle-invasive bladder cancer (FRART-BC study) [44]. Frailty was significantly associated with the trimodal therapy (TMT, TURBT, chemo- and radiotherapy) selection vs. radical cystectomy (RC), and overall survival was significantly shorter in the TMT group. It has also been demonstrated that frail patients are at the greatest risk for severe complications and mortality after RC [45].

Significant proportions of frail old patients are ineligible for cisplatin-containing chemotherapy, and carboplatin is used instead in this cohort. Carboplatin-based chemotherapy significantly decreases the likelihood of both OR and CR in patients with metastatic UC compared to cisplatin [46]. Thus, there is a need for effective and tolerable treatment options for these patients. In the long follow-up (up to 5 years), the KEYNOTE-052 study (first-line pembrolizumab in cisplatin-ineligible patients with advanced UC) demonstrated that pembrolizumab continued to elicit durable antitumor activity in cisplatin-ineligible patients with advanced UC [47].

## 7. Targeted Therapies in UC

Tyrosine-kinase inhibitors (TKIs), which are successfully used in metastatic renal cell carcinoma, have not demonstrated similar efficacy in BC. FGFR2 or FGFR3 alterations (mutation or fusion) are found in BC: although less than 10% of UC patients harbor an FGFR3 fusion, up to 60% have an FGFR mutation based on various reports. Erdafitinib, a TKI targeting fibroblast growth factor receptor (FGFR) has been approved by the FDA to treat BC. The efficacy of erdafitinib in BC was studied in a BCLC2001 phase II trial (Table 2, Figure 1). The patients involved in the study had advanced BC-bearing FGFR3 or FGFR2 genetic alterations. The overall response (OR) rate was 32.2%, and the complete response (CR) rate was 2.3%. After the study, erdafitinib was granted breakthrough therapy designation by the FDA to treat BC [48] (Figure 1, Table 3). Targeted therapies offer a potentially promising strategy for precision therapy in BC. Other FGFR inhibitors have the following response rate: 25.4% for infigratinib (phase III PROOF 302 trial) [49], 25% pemigatinib (phase I/II FIGHT-101 study) [50], and 24% for rogaratinib [51]. There are several trials of ICI-EGRF inhibitor combination treatment (NORSE, FORT-2, FIGHT-205).

## 8. Biomarkers for UC

Several trials have suggested that patients with a high expression of PD-L1 have a better response to PD-1/PD-L1 inhibition, while others have not demonstrated a similar correlation (reviewed in [52]). Thus, although PD-L1 testing is recommended to be routinely offered to patients with metastatic UC, PD-L1 expression is not a good predictive tool of ICI response in UC. So far, there are no reliable biomarkers for immunotherapy in UC. Although both nectin-4 and trop-a are widely expressed in metastatic UC, they have not been proved to be biomarkers for Enfortumab Vendotin and Sacituzumab Govitecan, respectively. There are no reliable biomarkers for ADC in UC.

In the absence of valuable biomarkers in BC, patient-derived organoids (PDO) could be a valuable tool to choose second and further lines of treatment. PDO represent a 3D culture, which mimics the biological characteristics of the primary tumors. In the first report on BC PDOs, they often retained parental tumor heterogeneity [53]. It has been demonstrated that urinary tract PDOs maintained inter-individual sensitivity towards targeting and cytotoxic agents [54], which is not always true for 2D primary culture.

## 9. DNA Damage Response Gene Alterations and Somatic Mutations in UC

UC has a high rate of somatic mutations (median 5.5/megabase) [55,56], placing it among the top 5 tumors with frequent somatic mutations alongside melanoma and lung cancer.

The exact gene or pathway alterations are as follows. Alteration of the p53 pathway was found in 89% of UC cases. The RTK/RAS/PI(3)K pathway was altered in 71% of cases. Alterations in DNA repair pathways included mutations in ATM (14%) and ERCC2 (9%) and deletions in RAD51B (2%) [9]. The increased somatic mutation burden (TMB) and generation of neoantigens in tumors theoretically might result in a more immunogenic tumor profile, suggesting a higher probability of response to immunotherapy. However, in the ABACUS trial studying PD-L1 inhibitor atezolizumab in a neoadjuvant setting, tumor mutational burden did not predict outcome [57].

Alterations in DNA damage repair (DDR) genes have been found in 2–14% of UC. PARP inhibitors are under evaluation in ongoing clinical trials in UC. In one study (NCT03397394), single agent rucaparib did not show significant activity in previously treated advanced UC patients regardless of HRD status. The enrollment was suspended at the first interim analysis [58]. An A031701 Phase II study examined the effects of neoadjuvant dose-dense gemcitabine and cisplatin (ddGC) in patients with MIBC. The presence of a DDR gene alteration was associated with chemosensitivity. None of those patients had experienced recurrence at a median follow-up of 2 years [59].

## 10. New Targets in UC

We have previously demonstrated that GSK-3 contributed to BC proliferation and survival. We showed that nuclear accumulation of GSK-3β is a novel prognostic marker in BC and identified GSK-3 as a potential therapeutic target in human bladder cancer [60].

We also demonstrated that 9-ING-41, a small molecule GSK-3 inhibitor, which is now in a phase 1/2 clinical study in patients with advanced cancers (https://clinicaltrials.gov/ct2/show/NCT03678883, accessed on 8 July 2022), induced cell cycle arrest, autophagy, and apoptosis in BC cells. The 9-ING-41 enhanced the growth inhibitory effects of gemcitabine or cisplatin, the autophagy inhibitor chloroquine, and sensitized BC cells to the cytotoxic effects of human immune effector cells [61].

Another potentially promising approach is targeting histone deacetylases (HDACs) with selective inhibitors. We have recently demonstrated that the inhibition of HDAC6 using selective specific small molecules could be a promising novel approach for the treatment of BC [62].

Various proteins as targets for cancer treatment are being extensively studied. Non-protein molecules could become novel potential targets for the treatment of BC. MicroRNAs (miRNAs) are small (containing about 22 nucleotides) single-stranded non-coding RNA molecules that exert post-transcriptional control of protein regulation. In cancer, miRNAs’ expression is deregulated, resulting in the elevated expression and activity of cancer-related proteins. Various miRNAs have been demonstrated to play a role in BC angiogenesis and metastasis. Identifying tumor-specific miRNA signatures could possibly promote the development of new markers as diagnostic and prognostic tools [63]. miRNA can be excreted into systemic circulation via exosomes. There is potential utility of exosomal miRNAs in serum or urine for a liquid biopsy of BC [64]. miRNA mimics, and molecules targeted at miRNAs (anti-miRs) could serve as therapeutic modalities in BC.

Long non-coding RNAs (LncRNAs) comprise a class of long (from 200 to nucleotides to 100 kilobases) RNAs that are not translated into protein. They have various cellular functions regulating gene expression via miRNA sponging, chromatin modification, transcriptional, and post-transcriptional processing [65]. Cancer-specific miRNAs can be detected in the serum and urine of cancer patients. LncRNAs could be useful as liquid biopsy markers of specific cancers. Using lncRNAs as therapeutic agents is a promising strategy in cancer treatment. One recent study [66] constructed an immune-related prognostic lncRNA signature using a bioinformatic approach. This was validated using RT-qPCR. An immune-related prognostic lncRNA signature, which consisted of RP11-89, PSORS1C3, LINC02672, and MIR100HG, might shed light on novel targets for individualized immunotherapy for BC patients.

## 11. Conclusions

The BC treatment therapy paradigm is changing rapidly. New agents such as ICI, targeted therapy, and antibody-drug conjugates have become hot issues in the treatment of advanced BC. Newly developed small molecules, antibodies, and combinatory treatment of various agents with a different mechanism of action are being tested in clinical trials. Novel approaches to develop reliable biomarkers need to select the patients who are most likely to benefit from each therapy. All these newly developed strategies predict opportunities for increased survival in BC.

## Figures and Tables

**Figure 1 jpm-12-01745-f001:**
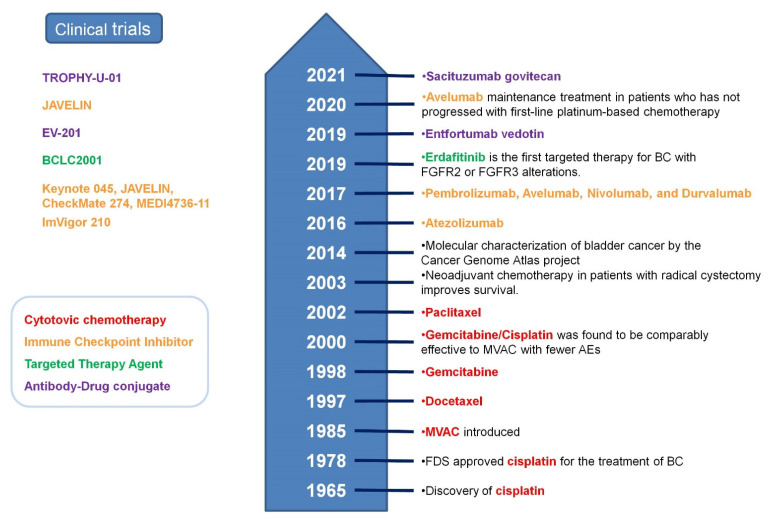
Timeline of BC clinical trials and FDA approval.

**Table 1 jpm-12-01745-t001:** Immuno-oncology drugs.

Target	Generic Name	Brand Name	Marketing Company
CTLA4	Ipilimumab	Yervoy	Bristol-Myers Squibb
PD-1	Nivolumab	Opdivo	Bristol-Myers Squibb (North America)
			Ono Pharmaceutical (other countries)
	Pembrolizumab	Keytruda	Merck Sharp & Dohme
PD-L1	Atezolizumab	Tecentriq	Genentech/Roche
	Avelumab	Bavencio	Merck KGaA and Pfizer
	Durvalumab	Imfinzi	Medimmune/AstraZeneca

**Table 2 jpm-12-01745-t002:** Key trials with ICIs, targeted therapies, and antibody-drug conjugates.

Name	Agent	Phase	Treatment Line	Target Patients	Number of Participants	Primary Outcomes
GO27831	Atezolizumab	Phase I		Patients with locally advanced or metastatic solid tumors	661	DLTs, MTD, RP2D, AEs
Checkmate 032	Nivolumab (as a single agent or in combination with ipilimumab)	Phase I	2nd line	Patients with advanced or metastatic solid tumors (6 tumor types including BC)	1131	ORR
Imvigor 210 Cohort 1	Atezolizumab	Phase II	1st line	Patients with locally advanced or metastatic urothelial BC. Treatment-naïve and ineligible for cisplatin-containing chemotherapy.	119	ORR of CR or PR
Imvigor 210 Cohort 2	Atezolizumab	Phase II	2nd line	Patients with BC who have progressed during or following a prior platinum-based chemotherapy regimen.	310	ORR
Keynote 052	Pembrolizumab	Phase II	1st line	Patients with advanced/unresectable or metastatic UC who are ineligible for cisplatin-based therapy.	374	ORR
Checkmate 275	Nivolumab	Phase II	2nd line	Patients with metastatic or unresectable BC who have progressed or recurred following platinum-based therapy.	386	ORR, ORR assessment by PD-L1 expression level
MEDI4736-1108	Durvalumab	Phase I/II	2nd line	Patients with advanced solid tumors.	1022	DLT in a dose-escalation phase. AEs in the Dose-escalation, Dose-exploration, and Dose-expansion Phase. ORR.
Keynote 012	Pembrolizumab	Phase IB	2nd line	Patients with advanced solid tumors, including advanced UC (Cohort C). Patients with PD-L1 expressing tumors were enrolled in Cohorts A, B, C and D.	297	ORR, AEs
JAVELIN	Avelumab	Phase I	2nd line	Various solid tumors including UC. Patients with metastatic or locally advanced solid tumors.	1756	DLT, BOR
Keynote 045	Pembrolizumab	Phase III	2nd line	Patients with metastatic or locally advanced/unresectable UC that has recurred or progressed following platinum-based chemotherapy.	542	PFS and OS in all Participants. PFS and OS in participants with PD-L1 Positive Tumors (CPS ≥ 1%). and strongly PD-L1 positive tumors (CPS ≥ 10%).
BCLC2001	Erdafitinib	Phase II	2nd line	Locally advanced and unresectable or metastatic urothelial carcinoma with prespecified FGFR alterations.	99	ORR
EV-201	Enfortumab vedotin	Phase II	2nd line	Locally advanced or metastatic UC who had previously treated with PD-1/PD-L1 inhibitor. Those who received prior treatment with platinum-containing chemotherapy were in cohort 1 and those who had no platinum-containing chemotherapy were in cohort 2.	219	ORR, CR, PR
EV-301	Enfortumab vedotin	Phase III	2nd, 3d line	Locally advanced or metastatic UC who have previously received platinum-based chemotherapy with a PD-1/PD-L1 inhibitor.	608	OS
TROPHY-U-01 cohort 1	Sacituzumab govitecan	Phase II	2nd line	Locally advanced or metastatic UC who have previously received a platinum-containing chemotherapy and a PD-1/PD-L1 inhibitor.	113	ORR, duration of response

DLT: Dose Limiting Toxicities; MTD: Maximum Tolerated Dose; RP2D: Recommended Phase 2 Dose; AEs: Adverse Events; ORR: Objective Response Rate; CR: Complete Response; PR: Partial Response; BOR: Best Overall Response; PFS: Progression Free Survival; OS: Overall Survival; CPS: Combined Positive Score (PD-L1); UC: urothelial cancer; BC: bladder cancer.

**Table 3 jpm-12-01745-t003:** FDA approved agents.

Date	Agent	Application
May 2016	Atezolizumab	2nd line
February 2017	Nivolumab	2nd line
April 2017	Atezolizumab	1st line for cis-ineligible patients
May 2017	Durvalumab	2nd line
	Avelumab	2nd line
	Pembrolizumab	1st line for cis-ineligible patients
		2nd line
April 2019	Erdafinib	1st line
December 2019	Ebfortumab vedotin	1st line
April 2021	Sacituzumab govitecan	1st line

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
