# Peer review of "Advanced Bladder Cancer: Changing the Treatment Landscape"

_jpm, 2022, doi:10.3390/jpm12101745_

Round 1
Reviewer 1 Report
It is a well-written narrative review, it does not add anything new to the literature and there is no detailed analysis, but it may be of global interest to readers.
Author Response
Thank you for you for your kind reply
Reviewer 2 Report
The manuscript by Bilim et al. summarized the treatments for advanced bladder cancer. This review could help readers understand the changes of the treatments for advanced bladder cancer and how the patients’ outcome has been improved. However, the writing needs to be improved and more detail needs to be added.
Comments:
1. I am bothered by the use of the abbreviations, like BC/BCa/BT. The authors properly defined BC the first time it appears in the article, but the BCa and BT are not defined. Do they both mean BC? If so, only one should be used to avoid confusion. Otherwise, the differences should be described. In addition, the term M-VAC on page 2 should be MVAC. What does QOL mean?
2. Line 46, “Novel classification based on genetic data has identified 5 unique molecular subtypes of BT…”. Do these subtypes need different treatments?
3. Line 54, “Despite currently available multimodality therapy, advanced bladder cancer has a very low overall survival (OS) rate of about 6%[9].” According to the data from https://www.cancer.net (“If the tumor is invasive but has not yet spread outside the bladder, the 5-year survival rate is 70%. About 33% of bladders cancers are diagnosed at this stage. If the cancer extends through the bladder to the surrounding tissue or has spread to nearby lymph nodes or organs, the 5-year survival rate is 38%. If the cancer has spread to distant parts of the body, the 5-year survival rate is 6%.”), the OS rates are different for different stages of advanced BC. These numbers should be provided in the article, otherwise it might mislead the readers.
4. To describe the phases of clinical trials, phase 1, 2,.. and phase I, II, … are used in different places. Although they are both acceptable, it should be consistent in one article.
5. The authors described a lot about urothelial cancer (UC). However, UC is the cancer from urothelial cells, and it not only exists in bladder cancer but also ureters, renal pelvis, and some other organs. Although UC is the most common type of BC, they are different. The authors should explain whether the treatments for UC are the same to those for BC/advanced BC or not.
6. A timeline of the development of the treatments would be very helpful for readers to have a clear global idea about this field.
7. Page 3, the authors also described “New options for the high-risk NMIBC”. Is the high-risk NMIBC considered as advanced BC? Do they use the same treatments?
8. Line 237, the information for the first reference is incomplete.
Author Response
We thank the reviewers for providing constructive comments and helping with the improvement of the manuscript quality. We have gone through each point carefully and the necessary answers are given below point by point. We believe that the manuscript is improved and is now suitable for publication.
Reviewer 2
The manuscript by Bilim et al. summarized the treatments for advanced bladder cancer. This review could help readers understand the changes of the treatments for advanced bladder cancer and how the patients’ outcome has been improved. However, the writing needs to be improved and more detail needs to be added.
Comments:
- I am bothered by the use of the abbreviations, like BC/BCa/BT. The authors properly defined BC the first time it appears in the article, but the BCa and BT are not defined. Do they both mean BC? If so, only one should be used to avoid confusion. Otherwise, the differences should be described. In addition, the term M-VAC on page 2 should be MVAC. What does QOL mean?
- ••Thank you for spotting these points. We changed BC/BCa/BT to BC through the manuscript. M-VAC on page 2 was changed to MVAC. QOL was spelled out (quality of life).
- Line 46, “Novel classification based on genetic data has identified 5 unique molecular subtypes of BT…”. Do these subtypes need different treatments?
- ••As per the reviewer’s suggestion, we added a discussion concerning different treatments based on bladder cancer molecular subtypes.
- Line 54, “Despite currently available multimodality therapy, advanced bladder cancer has a very low overall survival (OS) rate of about 6%[9].” According to the data from https://www.cancer.net (“If the tumor is invasive but has not yet spread outside the bladder, the 5-year survival rate is 70%. About 33% of bladders cancers are diagnosed at this stage. If the cancer extends through the bladder to the surrounding tissue or has spread to nearby lymph nodes or organs, the 5-year survival rate is 38%. If the cancer has spread to distant parts of the body, the 5-year survival rate is 6%.”), the OS rates are different for different stages of advanced BC. These numbers should be provided in the article, otherwise it might mislead the readers.
- ••Thank you for pointing this out. We added information to the article.
- To describe the phases of clinical trials, phase 1, 2,.. and phase I, II, … are used in different places. Although they are both acceptable, it should be consistent in one article.
- ••Thank you for spotting these points. Roman numerals are used throughout the manuscript to indicate clinical trials.
- The authors described a lot about urothelial cancer (UC). However, UC is the cancer from urothelial cells, and it not only exists in bladder cancer but also ureters, renal pelvis, and some other organs. Although UC is the most common type of BC, they are different. The authors should explain whether the treatments for UC are the same to those for BC/advanced BC or not. •••Thank you for pointing this out. We added information on upper urinary tract urothelial cancer to the manuscript.
- A timeline of the development of the treatments would be very helpful for readers to have a clear global idea about this field.
- ••As per the reviewer’s suggestion, we added Figure 1 presenting the timeline of approvals for advanced urothelial carcinoma.
- Page 3, the authors also described “New options for the high-risk NMIBC”. Is the high-risk NMIBC considered as advanced BC? Do they use the same treatments?
- ••Thank you for pointing this out. Although the high-risk NMIBC is not considered advanced BC, high-risk NMIBC progressing after BCG failure poses a serious clinical question as these tumors usually progress to advanced BC and need aggressive treatment like cystectomy. The primary response rate for intravesical BCG instillation is about 80%, but a durable response rate drops to only 30% after 10 years. Thus we added the pertinent information to the manuscript.
- Line 237, the information for the first reference is incomplete.
- ••Thank you for indicating this. The complete reference was added to the manuscript.
Reviewer 3 Report
This review article is concerned with the history, present and future of MIBC and treatment from the area of MVAC. I appreciate the review of the key clinical trials for advanced UC, while there are some comments as below,
1) Although the authors reviewed clinical trials for advanced UC and high risk NMIBC, what are the authors’ thoughts on the results of each? I am afraid that that simply summarizing the results of clinical trials can add little informative view or opinion in comparison to other reviews.
2) It would be helpful for readers to show a diagram/figure of changing treatment landscape.
3) It would also be desirable to describe as tables more in detail about the target patients, number of participants, primary outcomes, and results of key clinical trials that the authors introduced in the text.
4) There would be some differences of the treatment efficacy in UC subtypes, molecular signatures, tumor mutational burden, immune cells, etc., and it would be useful to provide information as "personalized medicine".
5) In addition to the targets identified by the authors, it would be beneficial to introduce other new targets.
Author Response
We thank the reviewers for providing constructive comments and helping with the improvement of the manuscript quality. We have gone through each point carefully and the necessary answers are given below point by point. We believe that the manuscript is improved and is now suitable for publication.
Reviewer 3
This review article is concerned with the history, present and future of MIBC and treatment from the area of MVAC. I appreciate the review of the key clinical trials for advanced UC, while there are some comments as below,
1) Although the authors reviewed clinical trials for advanced UC and high risk NMIBC, what are the authors’ thoughts on the results of each? I am afraid that that simply summarizing the results of clinical trials can add little informative view or opinion in comparison to other reviews.
- ••As per the reviewer’s suggestion, we added a concise discussion on each trial to the manuscript.
2) It would be helpful for readers to show a diagram/figure of changing treatment landscape.
- ••As per the reviewer’s suggestion, we added Figure 1 presenting a timeline of approvals for advanced urothelial carcinoma.
3) It would also be desirable to describe as tables more in detail about the target patients, number of participants, primary outcomes, and results of key clinical trials that the authors introduced in the text.
- ••As per the reviewer’s suggestion, we expanded Table 2 adding additional information on the clinical trials.
4) There would be some differences of the treatment efficacy in UC subtypes, molecular signatures, tumor mutational burden, immune cells, etc., and it would be useful to provide information as "personalized medicine".
- ••As mentioned in the review, up to date, there are no valuable biomarkers in BC. We extended the discussion of Keynote 045 trial which demonstrated a survival benefit in all subgroups with different levels of PD-L1 expression. Patient-derived organoids (PDO) could be a valuable tool for choosing second and further lines of treatment and could be useful in developing "tailored medicine".
5) In addition to the targets identified by the authors, it would be beneficial to introduce other new targets.
- ••As per the reviewer’s suggestion, we added novel potential targets for the treatment of BC. These include micro RNAs and long non-coding RNAs.
MicroRNAs (miRNAs)
Round 2
Reviewer 2 Report
The prior concerns have been properly addressed. Excellent job. I have no additional questions.
Reviewer 3 Report
The authors have made appropriate corrections and the manuscript appears to be valuable.